# Reagentless D-Tagatose Biosensors Based on the Oriented Immobilization of Fructose Dehydrogenase onto Coated Gold Nanoparticles- or Reduced Graphene Oxide-Modified Surfaces: Application in a Prototype Bioreactor

**DOI:** 10.3390/bios11110466

**Published:** 2021-11-19

**Authors:** Ieva Šakinytė, Marius Butkevičius, Vidutė Gurevičienė, Jonita Stankevičiūtė, Rolandas Meškys, Julija Razumienė

**Affiliations:** 1Department of Bioanalysis, Institute of Biochemistry, Life Sciences Center, Vilnius University, Saulėtekio av. 7, LT-10257 Vilnius, Lithuania; ieva.sakinyte@gmc.vu.lt (I.Š.); marius.butkevicius@gmc.vu.lt (M.B.); vidute.gureviciene@gmc.vu.lt (V.G.); 2Department of Molecular Microbiology and Biotechnology, Institute of Biochemistry, Life Sciences Center, Vilnius University, Saulėtekio av. 7, LT-10257 Vilnius, Lithuania; jonita.stankeviciute@bchi.vu.lt (J.S.); rolandas.meskys@bchi.vu.lt (R.M.)

**Keywords:** bioelectrocatalysis, Au nanoparticles, thermally reduced graphene oxide, direct electron transfer, biosensors, D-tagatose, fructose dehydrogenase, D-galactose bioconversion

## Abstract

As electrode nanomaterials, thermally reduced graphene oxide (TRGO) and modified gold nanoparticles (AuNPs) were used to design bioelectrocatalytic systems for reliable D-tagatose monitoring in a long-acting bioreactor where the valuable sweetener D-tagatose was enzymatically produced from a dairy by-product D-galactose. For this goal D-fructose dehydrogenase (FDH) from *Gluconobacter industrius* immobilized on these electrode nanomaterials by forming three amperometric biosensors: AuNPs coated with 4-mercaptobenzoic acid (AuNP/4-MBA/FDH) or AuNPs coated with 4-aminothiophenol (AuNP/PATP/FDH) monolayer, and a layer of TRGO on graphite (TRGO/FDH) were created. The immobilized FDH due to changes in conformation and spatial orientation onto proposed electrode surfaces catalyzes a direct D-tagatose oxidation reaction. The highest sensitivity for D-tagatose of 0.03 ± 0.002 μA mM^−1^cm^−2^ was achieved using TRGO/FDH. The TRGO/FDH was applied in a prototype bioreactor for the quantitative evaluation of bioconversion of D-galactose into D-tagatose by L-arabinose isomerase. The correlation coefficient between two independent analyses of the bioconversion mixture: spectrophotometric and by the biosensor was 0.9974. The investigation of selectivity showed that the biosensor was not active towards D-galactose as a substrate. Operational stability of the biosensor indicated that detection of D-tagatose could be performed during six hours without loss of sensitivity.

## 1. Introduction

A (bio)electrochemical detection and conversion of various chemical compounds is a rapidly evolving approach, which requires novel electrodes and optimized methods for immobilization of biocatalysts, particularly enzymes [1]. An efficient direct electron transfer (DET) from the enzymatic layer towards the electrode is a highly desired feature of an electrocatalytic system that allows development of a mediator-free approach, hence, enhancing the selectivity and sensitivity as well as reducing the costs of the analytic system [2,3] and creating a more efficient process, for which a lower impact on the environment might be expected [4]. The main trends to improve the electrode surface include: (i) chemical modification of the surface [5,6], (ii) enlargement of the electrode surface area by using various methods of chemical synthesis, etching or application of nanoparticles, and (iii) customization of the surface properties by other approaches [7]. The biocatalytic/sensing part of the system can be also improved by selection of an appropriate enzyme, and by genetic or chemical modifications directed to uniform immobilization of enzymes [8].

Graphene oxide- or graphite-based electrocatalytic systems attract attention due to many anticipated properties including morphology, surface chemistry and electrical characteristics [9]. Gold nanoparticles are also widely used for development of various biocatalytic processes [10].

Valorization of biomass by using electrochemical processes is becoming increasingly raised and numerous by-products are produced by conversion of agro-wastes. Hence, D-galactose can be obtained via lactose hydrolysis in the dairy industry [11]. Also, D-galactose is a by-product of the widely used third-generation ethanol production process from macroalgae [12,13]. Various attempts have been made globally to utilize this by-product by converting it to products with a higher additional value [11,14]. One among many products is a rare sugar D-tagatose. Since D-tagatose is very similar to the texture of sucrose and is 92% as sweet, but with only 7.3 kJ g^−1^ caloric value, which is 38% of the energy content of sucrose, it can be used as a natural low-calorie bulk sweetener [15,16,17]. In small quantities, D-tagatose can be naturally found in *Sterculia setigera* gum and various processed foods (sterilized and powdered cow’s milk, hot cocoa, and a variety of cheeses, yoghurts, and other dairy products), but its availability appears limited and its recovery is expensive, therefore creating a major impediment to its wider use in industry [17,18]. To overcome such unavailability, two different approaches including a chemical synthesis using a calcium catalyst [19] and a biochemical method using an L-arabinose isomerase as a biocatalyst [20,21,22] have been developed to produce D-tagatose from D-galactose. However, the chemical route has several disadvantages such as high temperature and high pressure during the process [23]. In contrast, recently significant literature has grown up around the theme of D-tagatose biosynthesis [24,25,26,27].

Obviously, evaluation of the total amount of D-tagatose in industrial products is very important for their quality; also it is very important for D-tagatose biosynthesis monitoring. At present, only several methods are suggested for D-tagatose determination: the colorimetric method [28], mass spectrometry [29] and gas chromatography [30,31]. Unfortunately, these methods do not allow easy and rapid monitoring, since they require relatively expensive instrumentation and well-trained operators; moreover, they often include a time-consuming sample pre-treatment step. Because of a high sensitivity, easier instrumentation, rapid (real-time) detection, low cost and ability to be used in turbid or fluorescent fluids, amperometric analysis offers a promising alternative to the conventional methods [32,33].

While D-tagatose is ketohexose and has a structure similar to D-fructose, except for the orientation of the hydroxyl group on C4 [34,35], D-fructose dehydrogenase from *Gluconobacter industrius* (FDH) was chosen and tested as a D-tagatose recognition element in this work. FDH catalyzes oxidation of D-fructose to 5-keto-D-fructose, but lately it has been shown that the FDH immobilized on nanoporous gold can weakly oxidize other sugars and polyols such as D-glucose, D-galactose and D-mannitol (the highest response of 5% was obtained with glucose) [36]. FDH is a flavohemoprotein, consisting of three subunits: subunit I, which is a catalytic domain containing a covalently bound FAD cofactor, where D-fructose is involved in a 2H^+^/2e^−^ oxidation to 5-keto-D-fructose; subunit II, which acts as a built-in electron acceptor with three heme c moieties covalently bound to the enzyme scaffold with two of them involved in the stepwise electron transfer pathway; subunit III, which is not involved in the electron transfer, but plays a key role for the enzyme complex stability [37,38]. A native FDH in bulk solutions does not catalyze oxidation of D-tagatose at all, however, recently it has been shown that FDH immobilized on the carbon paste electrodes modified with 2-arylamine-1,4-benzoquinone derivatives as electron transfer mediators can oxidize D-tagatose. Depending on the structure of the applied mediator, a signal reaches up to 30% of one observed in the presence of D-fructose [39]. Moreover, two additional benefits of this biocatalyst have been anticipated. First, the catalytic activity of FDH does not depend on oxygen [40]. Second, the FDH immobilized on the appropriate surfaces can direct shuttle electrons from the active center to the surface of the electrode [39,41]; hence, no additional components such as redox mediators are needed. Recently, it was demonstrated that thermally reduced graphene oxide (TRGO) and FDH, which were employed in the construction of a DET-based amperometric biosensor, may be used to measure D-fructose [42]. Several studies have also shown that modifying the electrode surface with various compounds (anionic or cationic) can be used to tune the enzyme’s selectivity. Changes in enzyme sensitivity and selectivity can be attributed to repulsion and/or attraction between the surface of the applied modified electrode and the enzyme’s amino acid side groups [43].

To combine a mediator-free DET approach with modulation of FDH substrate specificity via immobilization on different surfaces, three different electrode nanomaterials with immobilized FDH were used to design three types of biosensors: (i) monolayer of gold nanoparticles (AuNPs) coated with 4-mercaptobenzoic acid (4-MBA) (AuNP/4-MBA/FDH), (ii) AuNPs coated with 4-aminothiophenol (PATP) (AuNP/PATP/FDH) and (iii) a layer of TRGO on graphite (TRGO/FDH). The sensitivity and selectivity of those three systems to D-tagatose and their operational stability were investigated. The TRGO/FDH was also tested in a bioreactor that mimicked bioconversion technology in which by-product D-galactose was converted into D-tagatose by employing L-arabinose isomerase (L-AI).

## 2. Materials and Methods

### 2.1. Materials

D-Fructose dehydrogenase (EC 1.1.99.11) from *Gluconobacter industrius* (lyophilized powder; activity ≥ 20 U·mg^−1^ of solid) was purchased from Sigma-Aldrich (St. Louis, MO, USA). The recombinant thermophilic L-arabinose isomerase (L-AI) (5.6 U·mg^−1^) from *Geobacillus thermoleovorans* DSM 15325 was prepared as described previously [39]. The first fraction of thermally reduced graphene oxide (TRGO) was synthesized from the natural graphite according to the protocol reported by Šakinyte et al. [42]. Gold nanoparticles (AuNPs) were synthesized using HAuCl_4_⋅3H_2_O and trisodium citrate according to the Turkevich synthesis method [44]. The concentration of AuNP was calculated using the spectrophotometric method [45]. Graphite of extra pure grade was obtained from Merck (Darmstadt, Germany). Five hundred mM solutions of D-fructose, D-tagatose, and D-galactose were used in a McIlvaine buffer solution (pH 4.5) and in a 20 mM potassium phosphate buffer solution (pH 7.5) (PBS). Other chemical reagents of analytical grade were obtained from Sigma-Aldrich (Steinheim, Germany) and were used as arrived unless otherwise mentioned.

### 2.2. Enzyme Assay

L-AI activity was measured by determination of the amount of D-tagatose. Each reaction mixture contained 100 mM D-galactose and 1 mM MgCl_2_ in 50 mM PBS (pH 7.5). Unless stated otherwise, all reactions were performed at 50 °C for 5 min. The generated D-tagatose was determined by the cysteine carbazole sulfuric-acid method, and the absorbance was measured at 560 nm [46]. One unit of L-AI activity was defined as the amount of enzyme producing 1 μmol of D-tagatose per minute at 50 °C and pH 7.5. The concentration of protein was calculated following Bradford’s method using bovine serum albumin as the standard [47].

### 2.3. Preparation of Biosensors and Electrochemical Measurements

Aiming to design D-tagatose biosensor, TRGO was extruded by forming a tablet. The tablet was sealed in a Teflon tube with amorphous carbon pasta. Electrodes were washed with deionized water (DI), and dried before use. A biosensor was prepared by the adsorption on the TRGO surface of 2 µL, 0.5% triton X-100 solution in water (30 min, 10 °C) and 2 µL of FDH (1471 U·mL^−1^) in the McIlvaine buffer solution (pH 4.5) (30 min, at 10 °C). Then the biosensor was placed under a glutaro-aldehyde vapor condition at 20 °C for 30 min. Finally, the biosensor was designed by mechanically attaching and fixing the flexible terylene film with a rubber ring to the pretreated surface of the electrode. The basic scheme of the biosensor construction is presented in Figure 1.

Before the experiments, gold electrodes were polished with aluminum oxide slurry (0.3 μm), rinsed with deionized water and sonicated for 4 min in DI and 4 min in acetone to remove bounded particulates. After sonication, the working electrode was thoroughly washed with DI and treated by electrochemical cleaning. Briefly, 30 cyclic voltammetry (CV) scans were run from −0.2 to 1.75 V vs. Ag/AgCl and backwards in 0.5 M H_2_SO_4_, and the potential scan rate was 200 mV/s. Afterwards electrodes were thoroughly rinsed with deionized water and dried. Constructing the gold base D-tagatose biosensor, 3 μL of AuNPs (0.36 µM) were placed on the cleaned gold electrode surface and allowed drying at room temperature. Subsequently, the electrode was electrochemically cleaned (30 CV scans in 0.5 M H_2_SO_4_) and submerged in a 5 mM 4-mercaptobenzoic acid or 4-aminothiophenol solution in methanol and left overnight. Afterwards, the electrodes were thoroughly rinsed with deionized water and 2 μL of FDH (1471 U mL^−1^) in a McIlvaine buffer solution (pH 4.5) was placed and left for 30 min at 10 °C. The basic scheme of the biosensor construction is presented in Figure 1.

Amperometry measurements were performed using an electrochemical system (PARSTAT 2273, Princeton Applied Research, Oak Ridge, TN, USA) with a conventional three-electrode system comprised of a platinum plate electrode as an auxiliary, a saturated Ag/AgCl electrode as a reference and the working electrode D-tagatose biosensor designed on a base of three electrode surfaces. The response of the prepared biosensors to the addition of enzyme substrate was investigated under potentiostatic conditions at 0.4 V in a stirred McIlvaine, pH 4.5, and PBS, pH 7.5, buffer solutions. All measurements were obtained at 20 °C temperature.

From the experimental dependence of the current density (*j*) on substrate concentration (C) the apparent Michaelis constant values ($K_{M}^{app}$) and maximal current density (*j_max_*) were calculated. For this, the response current density was measured three times in the solution with C and the average response *j* was obtained. The experimental dependence *j* vs. C was approximated by OriginPro 8 (a free trial version from http://www.originlab.com, Origin Lab Corporation, Northampton, MA, USA; accessed on 13 September 2021) according to the electrochemical version of the Michaelis–Menten equation [48].

### 2.4. Enzymatic Synthesis of D-Tagatose

A prototype reactor for the synthesis of D-tagatose was designed as shown in Figure 2. L-AI from *G. thermoleovorans* DSM 15325 was used for bioconversion of D-galactose to D-tagatose. The production process of D-tagatose was carried out in a thermostatically isolated reactor (volume 9 mL) at 50 °C in a stirred PBS (pH = 7.5) containing 444.4 mM D-galactose. L-AI was kept in a dialysis bag in the center of the reactor (Figure 2).

During biosynthesis, the samples (150 µL) were taken every 10 h until 50 h to evaluate the concentrations of synthesized D-tagatose.

### 2.5. AFM Measurement 

TRGO and AuNP were analyzed by scanning probe microscope (D3100/Nanoscope IVa, Veeco Instruments Inc., Plainview, NY, USA). The tapping mode of surface scanning was used for visualization and characterization. The data and AFM images were processed by the NanoScope Software 6.14 (Veeco Instruments Inc.). The aqueous suspension of TRGO was obtained by mixing of 0.5 mm^3^ powder with 120 μL of distilled water. 10 μL of the suspension were dropped onto a silica plate and dried under 110 °C for 10 min. Then samples were left in a ventilating hood until the sample temperature decreased to 30 °C. Before each measurement the samples were additionally dried under 50 °C for 20 min. The aqueous suspension of the AuNPs were put on a gold disk electrode, which was cleaned as described previously and dried under a nitrogen stream at room temperature (RT). Imaging was performed in the air at RT.

## 3. Results and Discussion

### 3.1. Characterization of Electrode Surfaces

FDH can act as a DET-type enzyme in D-fructose bioelectrocatytic oxidation [42,49]; however, the efficiency of the DET reaction depends on various factors, which are related to both enzyme features and the structure of the electrode surface. Keeping in mind that a capability to oxidize D-tagatose also depends on an interaction between FDH and the electrode, what was already demonstrated in bioelectrocatalytic systems using different mediating materials [39], the key aspects of the oxidation of D-tagatose in a bioelectrocatalytic system operating on DET, would depend on the electrode material. Characterization of TRGO using X-ray diffraction, Raman spectroscopy, Brunauer–Emmett–Teller measurements, and elemental analysis has been performed in our previous work [42]. Here, the further examination of the TRGO- and AuNP-modified surfaces was carried out by using AFM. Two-dimensional representations of AFM topographic data of materials are shown in Figure 3A,B.

The AFM data clearly show that TRGO is composed of nanometric particles. The measured average diameter of the TRGO particles was ∼11 nm and the average particle height was ∼0.5 nm, which is very close to that of the single layer of graphene (0.39 nm).

Using AFM data, it was found that AuNPs increase the surface area by 50 percent. This difference was calculated by comparing the difference between the geometrically flat surface (2.25 μm^2^) and the measured surface area (3.55 μm^2^). In addition, AFM measurements revealed the size of AuNPs to be ∼19 nm; the size of AuNP was obtained using a ten-fold diluted nanoparticle solution. 

To confirm the DET, analysis of CVs obtained on AuNP/4-MBA, AuNP/PATP and TRGO with or without FDH was carried out. In fact, no increase in current was observed in CVs obtained on bare (without FDH) AuNP/4-MBA, nor for AuNP/PATP or TRGO after addition of D-tagatose or D-fructose (data not shown). In contrast, the CVs obtained on three types of electrodes with the enzyme exhibited a bioelectrocatalytic current (Figure 4). While the blank samples for AuNP/4-MBA/FDH, AuNP/PATP/FDH and TRGO/FDH electrodes showed no bioelectrocatalytic current in the McIlvain buffer, the addition of 10 mM D-fructose triggered the bioelectrocatalytic process on scanning from 0 to 0.55 V. Hence, it was concluded that a DET between the active center of FDH and surface took place. 

As can be seen in Figure 4 the capacitive current in CVs for the TRGO/FDH is much higher compared to CVs obtained for both bioelectrocatalyic systems using AuNP and FDH. This is, firstly, because the biosensor (TRGO/FDH) has an additional layer of semipermeable membrane, and secondly, due to functional groups on the surface of TRGO. However, in the CV of TRGO/FDH it is clearly seen that after addition of D-fructose at a potential of 0.4 V, which was later selected as the working electrode potential, the increase in current was several time higher compared to catalytic currents generated on AuNP-based electrodes. This can be explained by the fact that the thermal reduction procedure leads to formation of specific oxygen groups such as quinones, carboxy, lactone, epoxy, phenolic, and carbonyl that are capable to promote an electron and proton transfer on the surface of TRGO [42]. Due to oxygen-containing functional groups, the TRGO possesses the ability to transfer/receive electrons directly to/from enzymes, bypassing the need for an additional electron transfer mediator. Moreover, due to the large amount of these functional groups, the surface of TRGO becomes hydrophilic, which influences conformational changes of the immobilized enzyme, especially, when taking into account the hydrophobic nature of the heme c located inside FDH.

Previous studies [50,51,52] concluded that a proper orientation of the redox enzyme, such as FDH, on the electrode surface was of critical importance for successful direct electron transfer reactions of the enzyme on the electrode surface. Supposedly, the functional groups located on the surfaces of TRGO and AuNPs were able to take part in the reactions of electron transfer and also to position the FDH enzyme properly. 

### 3.2. Bioelectrocatalytic Properties of AuNP/4-MBA/FDH, AuNP/PATP/FDH and TRGO/FDH

Aiming to test the ability of FDH to catalyze the oxidation of D-tagatose to 5-keto-D-tagatose, FDH was immobilized onto electrode surfaces under experimental conditions and chronoamperometric measurements were performed. The responses of the manufactured biosensors to D-tagatose and D-fructose were recorded as a difference between the steady-state current and the background current. Conversion of both substrates by FDH immobilized on three tested electrode surfaces was observed. Taking into account the previous studies [37,38], it was assumed that the oxidation of D-tagatose occurred at the catalytic dehydrogenase domain, from which the electrons were transferred to the second subunit, the cytochrome domain containing the heme c, and finally shuttling the electrons to the electrode by generating an anodic current response directly proportional to the concentration of D-tagatose in the mixture. Hence, a DET between the active center of the FDH and surface took place. Dependences of steady-state current densities on D-tagatose and D-fructose concentrations are presented in Figure 5A,B.

The sensitivities of the biosensors were obtained from the slope of a linear relationship between current density and D-tagatose or D-fructose concentration presented in Figure 5. The values of sensitivities for bioelectrocatalytic oxidation of D-tagatose and D-fructose using different electrodes are presented in Table 1. Detailed analysis showed that the bioelectrocatalytic oxidation of D-tagatose on the proposed electrode surfaces was significantly lower comparing to using a common substrate—D-fructose. The maximal specificity to D-tagatose was obtained on the AuNP/PATP/FDH electrode and this value reached only 1.1%. The specificity for D-tagatose was calculated from the ratio *j_max_*(D-tagatose)/*j_max_*(D-fructose) ∗ 100% where *j_max_*(D-tagatose) and *j_max_*(D-fructose) are maximal current densities that can be generated by the bioelectrocatalytic system. *j_max_* for both substrates were theoretically calculated using calibration graphs (Figure 5) and the Michaelis–Menten equation. Varied specificities for all three biosensors (Table 1) demonstrated that FDH immobilization on these surfaces remained unique and resulted in slightly different conformations of FDH’s 3-dimensional shape.

According to data in Figure 5, the biosensors followed Michaelis–Menten kinetics. Thus, the apparent Michaelis constant values ($K_{M}^{app}$) were calculated using the electrochemical version of the Michaelis–Menten equation (Table 1). These values were higher comparing to the value of the native FDH obtained in solution (5 mM [53]) and indicated that the immobilization of FDH on all three surfaces complicated access or restricted binding of D-fructose to the active site of the enzyme [48], but at the same time, facilitated the access for D-tagatose. The most striking conclusion emerging from these data could be made that the FHD immobilized on the tested surfaces was active towards the D-tagatose, notwithstanding that the $K_{M}^{app}$ value for D-tagatose for all three biosensors was about eight-fold higher than that for D-fructose. Taking into account that native FDH did not catalyze oxidation of D-tagatose at all, it could be assumed that during immobilization the conformation of FDH was changed resulting in a proper spatial orientation, which was favored for an electrocatalytic oxidation of D-tagatose by FDH. 

The lowest value of $K_{M}^{app}$ for FDH, indicating a more “friendly” surface, was observed in the case with the enzyme operating onto TRGO (Table 1). The biosensor TRGO/FDH exhibited the best DET results in terms of an operational range (up to 40 mM), the highest sensitivity (0.03 μA mM^−1^ cm^−2^) and proper stability (data not shown).

In fact, the effective DET requires a proper spatial orientation of the enzyme, which should be located a short distance to the electrode surface in the way that the subunit II (the heme c-domain) of FDH should be facing toward the electrode surface. Recent research has shown that depending on the positive/negative surface charge, hydrophobicity/hydrophilicity determines the proper orientation of the enzyme [53], but this has only been demonstrated for subunit II of FDH and using D-fructose as the substrate. Our research showed that surface features have an impact on FDH selectivity, which we believe is linked to structural and conformational changes in the enzyme’s subunit I (the flavin-domain). The ionic and hydrophilic/hydrophobic interaction between the enzyme and the electrode surface causes changes in the orientation of the subunits as well as distortion in the FDH structure, particularly in subunit I. Due to these conformations, D-tagatose can access the active site of FDH and be oxidized there. Comparing to AuNP-based electrodes, the TRGO/FDH demonstrated the highest sensitivity and lowest specificity towards D-tagatose (Table 1). The surface of TRGO is the most hydrophilic due to the large amount of oxygen-containing functional groups what revealed TGA and elemental analysis. However, TRGO before immobilization of FDH was pretreated by triton X-100, so the hydrophilic polyethylene oxide chains were directed toward the surface of TRGO while the lipophilic aromatic hydrocarbon groups were directed toward the opposite site. Taking into account the hydrophobic nature of the heme c, the FDH enzyme must be properly situated on the surface pretreated with triton X-100. This is an assumption how the TRGO functional groups significantly influenced the distortion as well as accelerated the ET from the active site of the FDH to the electrode.

Since the biosensor TRGO/FDH exhibited the best DET results, it was decided to employ this biosensor for further research in the reactor for the bioconversion of D-galactose into D-tagatose.

### 3.3. Analysis of Stability and Selectivity of TRGO/FDH 

In order to study a possibility of the application of the biosensor in real media, a prototype bioreactor was designed (Figure 2). As the TRGO/FDH-based biosensor showed the highest sensitivity to D-tagatose, it was selected for D-tagatose monitoring in the bioreactor. To prevent surface fouling and ensure a prolonged operating, the biosensor was additionally coated by external semipermeable membrane (Figure 1). While bioconversion media contained a mixture of D-tagatose and D-galactose, it was necessary to investigate sensitivity of the biosensor to both carbohydrates and to compare with data obtained by alternative spectrophotometric analysis. In addition, the D-galactose bioconversion should be carried out at pH ~7.5; however, optimal pH for the native FDH has been determined around 4.5 [54]. To evaluate dependency of sensitivity of biosensor on pH, the amperometric current time responses to 4.4 mM D-fructose and D-tagatose in a McIlvaine buffer solution of pH 4.5 and in PBS of pH 7.5 were analyzed. A set of three TRGO/FDH biosensors designed in a same manner, was employed for the detection of D-fructose and D-tagatose at the same conditions. It was found that a residual activity of FDH towards D-fructose in PBS was about 37.84 ± 2.72% (the responses to D-fructose in the McIlvaine buffer solution was taken as 100%). Meanwhile, the response to D-tagatose in comparison with the response to D-fructose (which was taken as 100%) in PBS was 1.93 ± 0.47%. Hence, the specificity of the immobilized FDH towards D-tagatose also depended on the pH of the medium, probably due to changes of the enzyme’s spatial orientation.

The selectivity of the prepared biosensor was further studied to evaluate the influence of D-galactose on the determination of D-tagatose. Figure 6 shows amperometric current time responses of the D-tagatose biosensor in the presence of D-galactose added into the PBS.

As can be seen in Figure 6, no response was observed for the biosensor in the presence of different concentrations of D-galactose. Therefore, it could be concluded that D-galactose was not a substrate of FDH, and no interference due to D-galactose was observed for the tested biosensors. Meanwhile, the fast response of the biosensor towards D-tagatose could be achieved within hundreds of seconds after addition of D-tagatose. Furthermore, the biosensor response to D-tagatose did not change after adding of D-galactose.

Previously we showed that an amperometric biosensor based on TRGO and immobilized FDH displayed an appropriate long-term stability, hence, after a period of five days the biosensor sensitivity remained more than 80% of the initial response [42]. In this study the operational stability of the biosensors was inspected by measuring of 11 mM of D-tagatose solution in a stirred PBS (pH 7.5) (Figure 7).

As can be seen in Figure 7, six D-tagatose assays performed over a 6 h period were without any marked loss of sensitivity of the biosensor. On a basis of this assessment, a relative standard deviation (CV) of 0.34% for D-tagatose assays was obtained. These results suggested good reproducibility and operational stability of the developed biosensor. Thus, properties of the THGO/FDH biosensor allowed us to monitor D-tagatose levels in real samples collected during the D-galactose bioconversion reaction.

### 3.4. Application of the Biosensor in D-galactose Bioconversion Reactor

Aiming to demonstrate practical applicability of the proposed biosensors, the TRGO/FDH biosensor was employed for the quantification of D-tagatose in the prototype bioreactor (Figure 2). This reactor demonstrated the technological possibility of converting the by-product D-galactose into a promising sweetener D-tagatose. To evaluate the applicability of the biosensor, the single standard addition method was applied [55]. The method of single standard addition involves measuring the current time response for the reaction mixture samples with unknown D-tagatose content, and then measuring the current time response of a sample to which a known amount of analyte (11 mM of D-tagatose) was added. Thus, two measurements were undertaken for calculation of the D-tagatose concentration in a given bioconversion reaction mixture: before the addition of the standard and after the addition of the standard. The D-tagatose amounts obtained in the bioconversion reaction mixture using the method described above are summarized in Table 2. The average values of D-tagatose and their subsequent associated standard deviations were calculated using three independent measurements. The same samples of reaction mixture in terms of D-tagatose amount were also analyzed by the alternative spectrophotometric method. The results of spectrophotometric analysis compared with those obtained by TRGO/FDH biosensor are presented in Table 2.

Based on data presented in the Table 2, it was estimated that bioconversion reaction yield of 21% was achieved after 50 h.

The accuracy of the amperometric biosensor was confirmed by plotting the results obtained by the amperometric biosensor vs. the results obtained using spectrophotometric analysis (Figure 8).

The correlation coefficient (r_xy_) between two independent analyses: spectrophotometric and by the TRGO/FDH biosensor of the bioconversion mixture was 0.9974. This indicated an excellent agreement between the two methods. The slope of the correlation straight was of 0.9978, which indicates, that results, determined using the amperometric biosensor, were slightly lower than those of spectrophotometric analysis. These findings suggest that, in general, the TRGO/FDH biosensor generated a correct response to D-tagatose in the D-galactose/D-tagatose mixture.

These experiments confirmed that the designed biosensor could be used for D-tagatose monitoring in such type of bioreactors as well as being promising for future food technologies.

## 4. Conclusions

A bioelectrocatalytic oxidation of D-tagatose based on a direct electron transfer was observed using immobilized FDH on three different electrode surfaces: gold nanoparticles (AuNPs) coated with 4-mercaptobenzoic acid (AuNP/4-MBA) or 4-aminothiophenol (AuNP/PATP) monolayer, and a layer of thermally reduced graphene oxide (TRGO) on graphite.

Because native FDH does not catalyze D-tagatose oxidation in bulk solutions, it can be concluded that oriented immobilization of the enzyme onto the proposed electrode surfaces modifies FDH’s selectivity towards D-tagatose. Different values of specificity to D-tagatose for all three biosensors revealed that immobilization of FDH on these surfaces remained unique and herewith led to slightly different conformations of the 3-dimensional form of FDH. Notwithstanding that a bioelectrochemical response towards D-tagatose is significantly lower comparing to the response to common substrate—D-fructose, the developed biosensors are entirely applicable to monitor a formation of D-tagatose during the isomerization process of D-galactose.

This research demonstrated that a specificity of FDH to D-tagatose can be changed using proper electrode materials for immobilization of the enzyme. We propose that during immobilization, FDH undergoes conformational changes as it binds to the electrode surface, resulting in the proper spatial orientation required for direct D-tagatose electrocatalytic oxidation. This assumption was confirmed by calculated values of the apparent Michaelis constant ($K_{M}^{app}$), which were higher in comparison to the value of the native FDH obtained in solution and varied from 65 to 210 mM, depending on the electrochemical platform used for immobilization of FDH.

As TRGO/FDH demonstrated the highest sensitivity to D-tagatose, it was chosen to investigate the biosensors’ applicability in a prototype bioreactor. Independent spectrophotometric analysis of the bioconversion mixture revealed that the biosensor was suitable for monitoring of D-tagatose during a bioconversion of D-galactose by L-arabinose isomerase. Because the biosensor showed no response in the presence of various quantities of D-galactose, it may be concluded that D-galactose is not a substrate for FDH and hence has no effect on the biosensor’s response.

While a simple approach for monitoring D-tagatose in industrial projects is still in great demand, the proposed electrochemical biosensors could address that void. Because there is currently no enzyme for selective D-tagatose oxidation that could serve as a recognition element for electrochemical biosensors, we propose an alternative—FDH with adjusted D-tagatose selectivity. Such a biotechnological solution could be very promising for development into a process for valorizing dairy industry waste.

## Figures and Tables

**Figure 1 biosensors-11-00466-f001:**
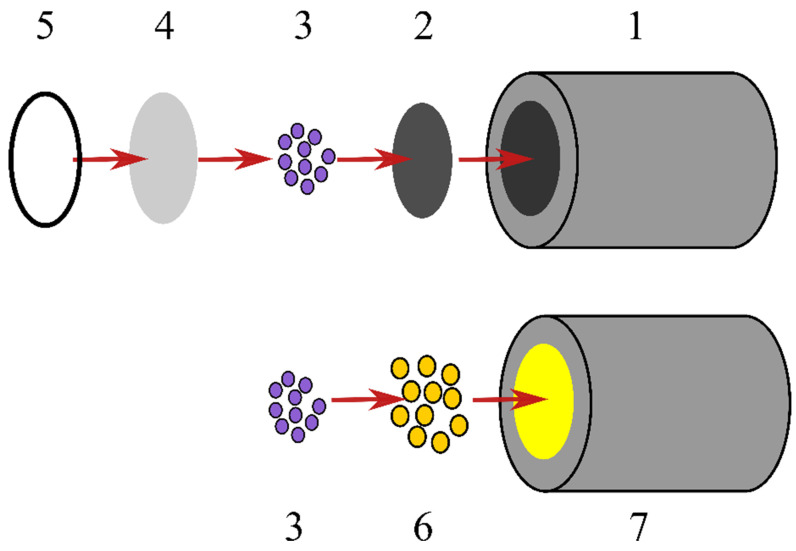
Basic scheme of the D-tagatose biosensor: 1—graphite electrode, 2—layer of TRGO, 3—layer of FDH, 4—terylene film, 5—rubber ring, 6—AuNPs modified with 4-mercaptomenzoic acid or 4-aminothiophenol, and 7—gold electrode.

**Figure 2 biosensors-11-00466-f002:**
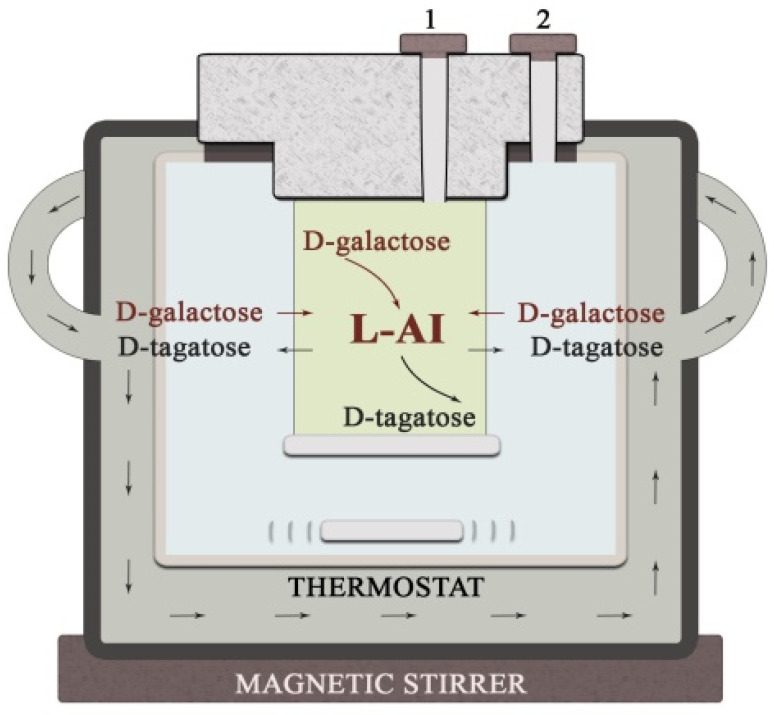
A prototype bio reactor for the production of D-tagatose. 1—L-AI entry channel, 2—sampling channel for the monitoring of the conversion progress.

**Figure 3 biosensors-11-00466-f003:**
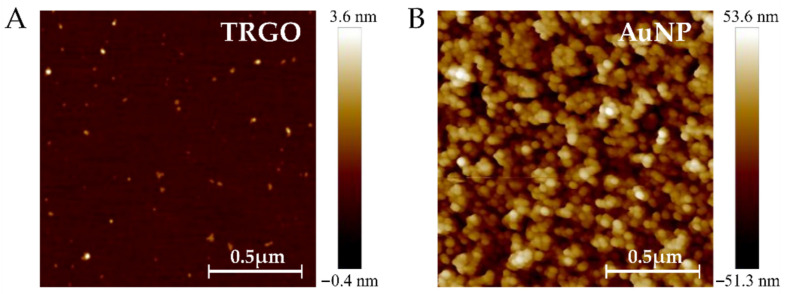
Two-dimensional AFM images of the TRGO deposited on the silica plate (**A**) and AuNP deposited on flat gold (**B**). AFM images were obtained by the tapping mode in air.

**Figure 4 biosensors-11-00466-f004:**
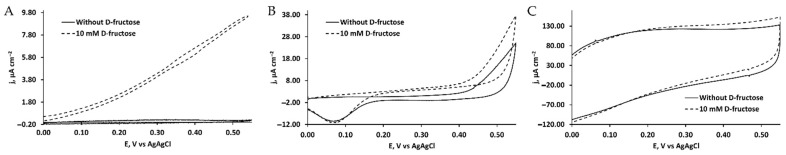
Cyclic voltammograms of AuNP/4-MBA/FDH (**A**), AuNP/PATP/FDH (**B**), and TRGO/FDH (**C**). Black curve—in the absence of D-fructose, and dashed curve in the presence of 10 mM D-fructose. McIlvaine buffer solution, pH 4.5, 20 °C, with a scan rate of 10 mV s^−1^.

**Figure 5 biosensors-11-00466-f005:**
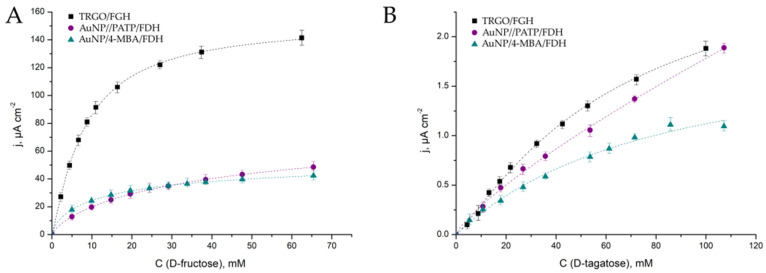
Calibration graphs for three types of sensors for changing concentrations of D-fructose (**A**) or D-tagatose (**B**). Concentrations measured under potentiostatic conditions at 0.4 V vs. Ag/AgCl in a stirred McIlvaine buffer solution, pH 4.5, 20 °C. The experimental data are fitted by using the Michaelis–Menten equation.

**Figure 6 biosensors-11-00466-f006:**
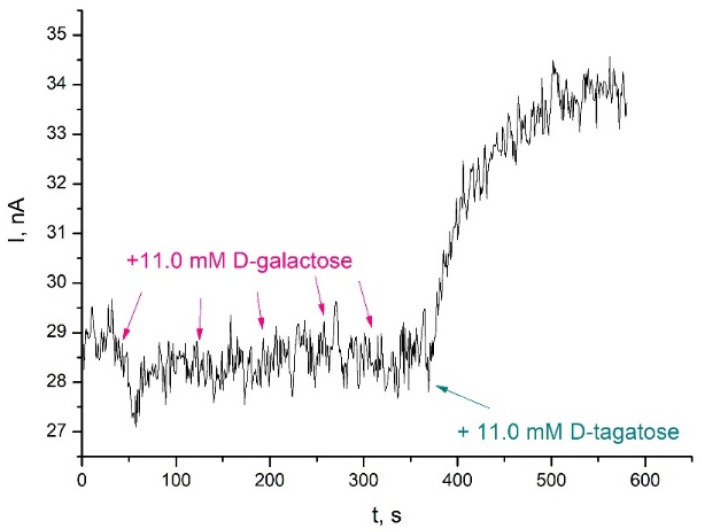
TRGO/FDH responses to D-galactose and D-tagatose. Measurement was performed in a stirred McIlvaine buffer solution, pH 4.5, 20 °C, under potentiostatic conditions (0.4 V vs. Ag/AgCl).

**Figure 7 biosensors-11-00466-f007:**
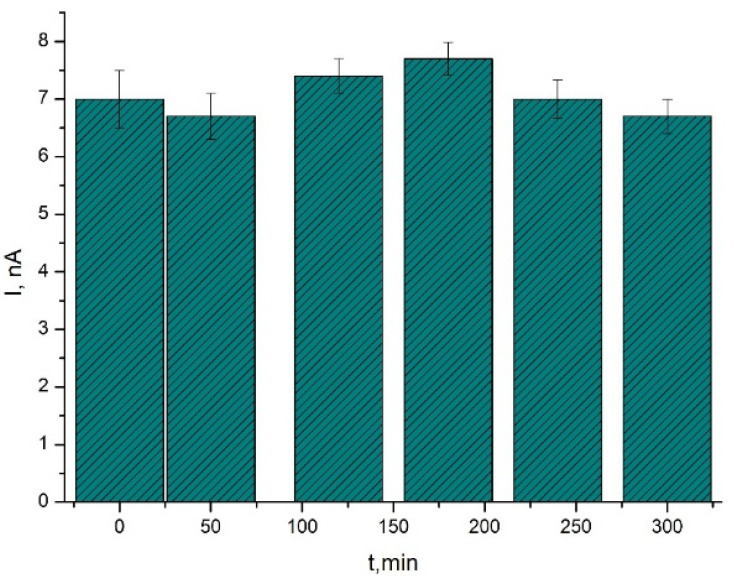
Reproducibility and operational stability of the biosensor. D-Tagatose (11.0 mM) in a stirred PBS, pH 7.5.

**Figure 8 biosensors-11-00466-f008:**
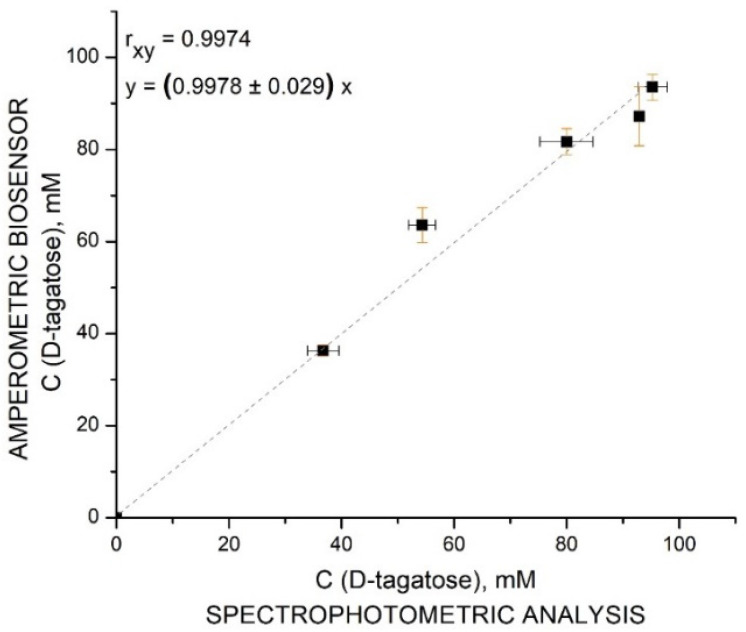
Correlation between the analysis of the formed D-tagatose using the TRGO/FDH biosensor and spectrophotometric assay in the bioreactor during bioconversion of D-galactose.

**Table 1 biosensors-11-00466-t001:** Main parameters of bioelectrocatalytic oxidation of D-tagatose and D-fructose using proposed biosensors.

	TRGO/FDH	AuNP/4-MBA/FDH	AuNP/PATF/FDH
Liner range (D-tagatose), mM	4.4 *–32.3	5.4 *–19.3	5.4 *–29.5
Sensitivity (D-tagatose), μA/mMcm^2^	0.030 ± 0.002	0.019 ± 0.002	0.025 ± 0.001
$K_{M}^{app}$ (D-fructose), mM	8.1 ± 0.2	9.9 ± 0.6	24.8 ± 1.5
$K_{M}^{app}$ (D-tagatose), mM	65 ± 10	86 ± 13	210 ± 20
Specificity (D-tagatose),%	0.33 ± 0.08	0.68 ± 0.09	1.1 ± 0.1

* The lowest measured concentration.

**Table 2 biosensors-11-00466-t002:** Comparison of D-tagatose concentrations formed during isomerization of D-galactose obtained using two methods: amperometric biosensor vs. spectrophotometric.

Duration of Bioconversion, h	D-Tagatose Formed, mM	
	Amperometric Biosensor	Spectrophotometric Analysis
0	0.0	0.0
10	36.3 ± 1.2	36.7 ± 2.8
20	63.6 ± 3.8	54.3 ± 2.4
30	81.7 ± 2.8	80.0 ± 4.7
40	87.2 ± 6.4	92.9 ± 6.5
50	93.5 ± 2.9	95.3 ± 2.6

## Data Availability

All data is contained within the article.

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
