# Peer review of "Reagentless D-Tagatose Biosensors Based on the Oriented Immobilization of Fructose Dehydrogenase onto Coated Gold Nanoparticles- or Reduced Graphene Oxide-Modified Surfaces: Application in a Prototype Bioreactor"

_biosensors, 2021, doi:10.3390/bios11110466_

Round 1

Reviewer 1 Report

The manuscript describes reagentless D-tagatose biosensors based on the oriented im mobilization of fructose dehydrogenase onto coated gold nanoparticles or reduced graphene oxide-modified surfaces. The manuscript has been well organized. I has some comments.

  1. What is the process of bioelectrocatalytic? Do different electrochemical methods: cyclic voltammetry and chronoamperometry have an impact on the bioelectrocatalytic process?
  2. Three different biosensors can immobilize fructose dehydrogenase on the electrode surface. What is the main force to immobilize it, and what is the reason why it can orient the fructose dehydrogenase?
  3. Under what conditions is the single standard addition method suitable for quantitative detection? Why choose 11 mm D-tagatose as the concentration unit of single addition?
  4. What is the significance of using biosensors to detect the production of D-tagatose in a prototype bioreactor?
  5. Why choose PBS buffer instead of McIlvaine buffer in the detection application of prototype bioreactor?

Reviewer 2 Report

This manuscript reports on the “Reagentless D-tagatose biosensors based on the oriented immobilization of fructose dehydrogenase onto coated gold nanoparticles- or reduced graphene oxide-modified surfaces: Application in a prototype bioreactor.” The content of the work is interesting, I recommend the publication of the manuscript in its present form to the Biosensors.

Reviewer 3 Report

Sakynite and co-workers report the development of biosensors for detection of D-tagatose based on immobilized fructose dehydrogenase. The electrochemical response of the enzyme towards both D-fructose and D-tagatose is characterized on different surfaces, namely AuNPs coated with 4-mercaptobenzoic acid or 4-aminothiophenol, and reduced graphene oxide. The results show that the immobilized FDH displays a low D-tagatose oxidation activity on all surfaces in contrast to its behavior in solution. The graphene oxide based biosensor was subsequently employed for quantification of D-tagatose in a D-galactose converting bioreactor. The results are interesting, however the following issues should be addressed.

1) Throughout the manuscript, including in the title, the authors state that the ability of FDH to catalyze the oxidation of D-tagatose is due to changes in conformation of the enzyme upon immobilization and favorable spatial orientation on the electrode that lead to an altered enzyme selectivity. The different catalytic response obtained for the three immobilization methods is also attributed to slightly different conformations of the immobilized FDH. However, no direct evidence is presented in the paper to support these claims, which seem to be based solely on the catalytic response of the bioelectrodes. This should be clearly addressed in the paper.

2) The characteristics of the electrodes used for immobilization of FDH are mentioned in the manuscript, such as the charge of the monolayers coating the AuNPs, the functional groups of TRGO and electrode surface areas, however these are not taken into consideration in the discussion of results. The type of interactions that can potentially be established between FDH and the different electrode surfaces should also be addressed in the manuscript.

3) It is not clear if the dialysis membrane is always used in the construction of the TRGO/FDH (as indicated in Materials and Methods section and Figure 1) or only when measuring D-tagatose on the bioreactor set-up, as suggested in section 3.3 (line 319-320). How does the dialysis membrane affect the performance of the biosensor?

4) The involvement of the TRGO functional groups in the ET reactions (line 257) should be explained in detail.

5) The CVs represented on Figure 4 show much clearer catalytic response to D-fructose for the AuNP/4-MBA/FDH electrode than for the TRGO/FDH system, owing to the high capacitive currents observed in the later case. The authors should discuss the differences between the capacitive currents measured with each electrode system under study and how this could affect the sensitivity of the resulting biosensors.

6) Complete captions should be provided for Figures 3 and 6.

7) Results on Figure 7 show that the current response is stable for 6 h, however the biosensor was used up to 50 h in the bioreactor. Is the device stable for the duration of the assay in the bioreactor?

Other issues:

  1. The 1st paragraph in the introduction is too general, more specific details should be provided. Line 42-43, although the selection of an appropriate electrode material is very important for biosensor design, it cannot be considered as a way to improve a particular electrode surface which is the topic of the sentence; line 46 more information must be provided about other approaches that can be used to customize electrode surface properties.
  2. Line 50 – It is not clear what “anticipated properties” and “electrical characteristic” of graphene and graphite materials means. The advantages of using this type of materials as electrodes are widely recognized, including the conductive properties.
  3. Line 109-110 – Sentence should be re-written for clarity. All amino acids are ionic.
  4. Line 161 – The concentration of AuNPs used for the preparation of the biosensors should be reported in materials and methods section.
  5. Line 215-217 – The sentence seems to repeat what is stated in the previous sentence.
  6. Line 233-234 – The authors should explain why the charge of the monolayers coating the AuNPs is relevant for the discussion of AFM results.
  7. Lines 253-257/258-251 – The two sentences seem to repeat the same idea.
  8. As reported on line 288 and Table 1 the sensitivity and current density ratios, of TRGO/FDH response to D-tagatose vs D-fructose appear to be identical, 1.1%. To avoid confusion, the sensitivity (or the sensitivity ratios) of the three biosensors to D-fructose could be added to Table 1. The concentration at which the j(D-tagatose)/j(D-fructose) ratios were determined should also be indicated in the manuscript.
  9. Line 371-372 – Why was the standard addition method employed? Are the calibration parameters of the biosensor different in the bioreactor and in the electrochemical cell? This should be discussed.
  10. Line 406-409 – The enzyme should be mentioned in the 1st paragraph of the conclusion.
  11. Line 424-426 – The reasons for comparing FDH’s KM value for fructose in solution with the KMapp for D-tagatose with immobilized FDH are not clear. The access of substrates to the active site of the immobilized enzyme should also be considered.

Round 2

Reviewer 1 Report

The manuscript has been well modified and can be accepted in its current state.

Author Response

Thank you for your review.

Reviewer 3 Report

The authors have made important changes to the manuscript. However some issues have not been addressed in the revised version. The manuscript could be suitable for publication provided further improvements are added.

Line 42-43 – The sentence was not changed as indicated in the authors response file (rev 3 point 8), therefore the problem persists: although the selection of an appropriate electrode material is very important for biosensor design, it cannot be considered as a way to improve a particular electrode surface which is the topic of the sentence.

Line 260-263 – The sentence appears to contradict work published in the literature, including the cited reference. Thermal reduction of graphene oxide typically results in the removal of oxygen-containing functional groups, restoring the conductivity of graphene oxide by reinstating double-bonded aromatic carbon atoms, while still retaining some chemically active oxygen groups that can facilitate e.g. interactions with biomolecules.

Line 265-268 – Why does the hydrophilic character of TRGO influence the internal properties of FDH, particularly the expected hydrophobic environment surrounding the heme group that is buried inside the protein chain? This should be explained.

Line 352-353 – The authors should clarify that the dialysis membrane is an integral part of the biosensor and not only added during measurements of D-tagatose in the bioreactor set-up. Consider removing the word “additionally”.

Figure 6 – Figure caption should be self-sustaining. Considering three types of biosensors were developed the authors should indicate which system the data refers to.
